# Merging without Forgetting: Continual Fusion of Task-Specific Models via Optimal Transport

## Abstract

Merging models fine-tuned for different tasks into a single unified model has become an increasingly important direction for building versatile, efficient multi-task systems. Existing approaches predominantly rely on parameter interpolation in weight space, which we show introduces significant distribution shift in the feature space and undermines task-specific knowledge. In this paper, we propose OTMF (Optimal Transport-based Masked Fusion), a novel model merging framework rooted in optimal transport theory to address the distribution shift that arises from naive parameter interpolation. Instead of directly aggregating features or weights, OTMF aligns the semantic geometry of task-specific models by discovering common masks applied to task vectors through optimal transport plans. These masks selectively extract transferable and task-agnostic components while preserving the unique structural identities of each task. To ensure scalability in real-world settings, OTMF further supports a continual fusion paradigm that incrementally integrates each new task vector without revisiting previous ones, maintaining a bounded memory footprint and enabling efficient fusion across a growing number of tasks. We conduct comprehensive experiments on multiple vision and language benchmarks, and results show that OTMF achieves state-of-the-art performance in terms of both accuracy and efficiency. These findings highlight the practical and theoretical value of our approach to model merging.

## 1 Introduction

Large-scale pretrained models (PTMs) have achieved remarkable success across natural language processing, computer vision, and multimodal understanding (Bommasani et al., 2021; Radford et al., 2021). As their adoption accelerates, integrating multiple fine-tuned models into a unified multi-task system has become a key challenge (Yang et al., 2024e; Tang et al., 2024a). Traditional multi-task learning (MTL) methods rely on joint training with shared representations (Misra et al., 2016; Sener & Koltun, 2018; Ma et al., 2018), but such approaches are often infeasible when data access is restricted by privacy, communication, or resource constraints (Wortsman et al., 2022a; Li et al., 2023; Wu et al., 2024).

To overcome these limitations, *model merging* has emerged as a promising alternative, enabling the construction of unified models by directly combining independently fine-tuned task models (Wortsman et al., 2022b; Yang et al., 2024a; Zhou et al., 2024).Existing methods include *Weight Averaging*, Fisher-weighted averaging (Wortsman et al., 2022b; Matena & Raffel, 2022b), *Task Arithmetic*(Ilharco et al., 2022; Ortiz-Jimenez et al., 2024), and *Ties-Merging*(Yadav et al., 2023; Davari & Belilovsky, 2023), often relying on the assumption of *mode connectivity*—i.e., smooth paths exist between optima in parameter space (Garipov et al., 2018; Draxler et al., 2018).

However, most of these methods operate solely at the parameter level, assuming linear interpolation can preserve task knowledge. In practice, they often disrupt feature distributions, leading to degraded performance in heterogeneous settings (Ilharco et al., 2022; Yadav et al., 2023). This raises a key question:

*How can we merge models while preserving the distributional structure of each task and promoting cross-task knowledge integration?*

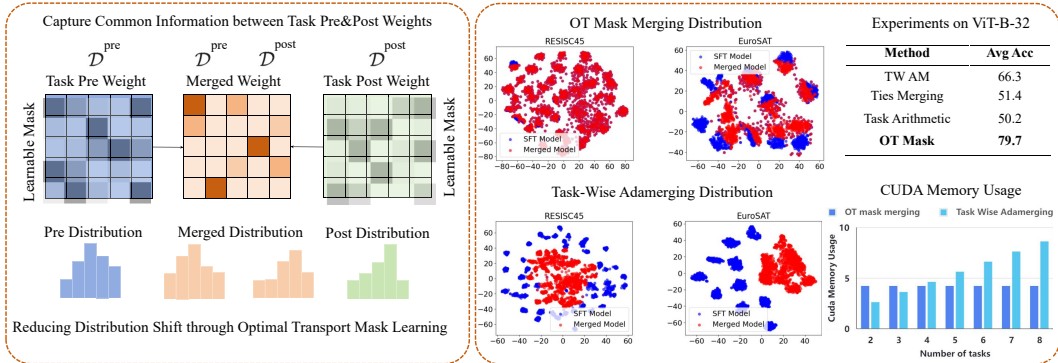

Figure 1: **Left:** OTMF captures common information between pre/post weights while reducing distribution shift. **Middle:** T-SNE visualizations show that OTMF yields output distributions closely aligned with the pre model's distributions, outperforming Task-wise AdaMerging. **Right:** OTMF outperforms other sequential methods in average accuracy while using less CUDA memory than Task-Wise AdaMerging, highlighting its advantages in both performance and efficiency.

We argue that prior methods fail to maintain the semantic geometry of task-specific feature spaces. Instead of directly editing parameters, we propose a principled solution grounded in optimal transport: aligning latent distributions to extract shared representations. Specifically, we introduce the **Optimal Transport-based Masked Fusion (OTMF)** framework (Figure 1), which derives common masks via optimal transport plans between model distributions. These masks selectively preserve task-agnostic components while minimizing distribution shift (Singh & Jaggi, 2020; Ortiz-Jimenez et al., 2023).

Furthermore, OTMF supports a memory-efficient **continual merging** paradigm: at each step, only the current merged model and the incoming task model are required, resulting in a fixed memory footprint that does not grow with the number of tasks (Figure 1, right). This makes OTMF well-suited for scalable multi-task and continual fusion (Wang et al., 2023; Wu et al., 2023).

As illustrated in Figure 1, our method reduces distribution shift via latent alignment (left) and achieves better clustering in T-SNE visualizations (middle). Empirically, OTMF consistently outperforms baselines on ViT-B/32 benchmarks (Wortsman et al., 2022b; Yang et al., 2024c), while maintaining low memory cost in sequential settings.

**Our main contributions are:**

1. We propose the **Optimal Transport-based Masked Fusion (OTMF)** framework, which explicitly aligns task-specific distributions in feature space, addressing key limitations of parameter-based merging.

2. We design a scalable, memory-efficient **continual merging** paradigm that incrementally integrates tasks with constant memory, suitable for large-scale deployment.

3. We conduct comprehensive experiments across vision and language tasks, demonstrating OTMF's superiority in accuracy, efficiency, and robustness over state-of-the-art baselines.

## 2 RELATED WORK

**Model Merging and Continual Fusion.** Model merging integrates independently fine-tuned models into a unified one without accessing the original training data (Tang et al., 2024a), differing from ensembling (Sagi & Rokach, 2018; Arpit et al., 2022; Liu & Soatto, 2023) and model mixing (Yadav et al., 2024; Komatsuzaki et al., 2022) by preserving the original architecture and avoiding extra inference cost. A dominant strategy is weight interpolation based on linear mode connectivity (Frankle et al., 2020; Draxler et al., 2018; Izmailov et al., 2018), which has proven effective in applications like LLM alignment (Lin et al., 2024; Chegini et al., 2024; Gorbatovski et al., 2024; Pentyala et al., 2024), auxiliary-task learning (Jiang et al., 2023), and multi-objective optimization (Rame et al., 2024; Chen & Kwok, 2024; Tang et al., 2024b). In contrast, alignment-based methods address inter-model discrepancies through permutation matching (Ainsworth et al., 2022; Li et al., 2024b), channel-wise

alignment (Liu et al., 2022), or activation reordering (Stoica et al., 2023; Jin et al., 2022; Kinderman et al., 2024; Xu et al., 2024), though they typically require simultaneous access to all models, limiting scalability.

To overcome this, continual model merging incrementally integrates task-specific models without revisiting earlier data. Initial approaches rely on parameter averaging or task vector arithmetic (Ilharco et al., 2022), while recent work enhances robustness through subspace preservation (Tang et al., 2023), dynamic routing (Li et al., 2024a), and contrastive alignment (Yang et al., 2024b). Our method builds on this line by introducing a scalable fusion paradigm based on optimal transport and selective mask updates, enabling efficient and memory-bounded continual integration.

**Optimal Transport for Distribution Alignment.** Optimal Transport (OT) provides a theoretical basis for comparing distributions by minimizing transport cost (Monge, 1781; Kantorovich, 1942), with scalable versions such as Sinkhorn regularization (Cuturi, 2013; Cuturi & Doucet, 2014). OT has proven effective in domain adaptation (Courty et al., 2016; Xu et al., 2020), generative modeling (Bousquet et al., 2017), and distributional robustness (Lu et al., 2023). For model fusion, OT has been applied to neuron matching (Li et al., 2016; Yurochkin et al., 2019), extended to CNNs in federated contexts (Wang et al., 2020).Our approach differs fundamentally from prior works that use optimal transport for permutation alignment. In continual merging setting, we compute an OT loss between the merged model and the task-specific models to explicitly reduce residual distribution shift. By leveraging learnable masks, OTMF performs distribution-level alignment in feature space—rather than mere parameter-space regularization—thereby mitigating task interference and better preserving semantic structures and model performance during fusion.

## 3 PRELIMINARIES

This section introduces the theoretical foundations for the OTMF framework. We begin by formulating a representation-level measure of **distribution shift**, then introduce an **optimal transport**–based formulation for alignment, and finally formalize the **continual fusion** process.

### 3.1 PROBLEM FORMULATION: DISTRIBUTION SHIFT

Let $x \in \mathcal{X}$ denote input samples, and consider two fine-tuned models $\theta_{\mathrm{pre}}$ and $\theta_{\mathrm{post}}$ producing latent features $f_{\theta_{\mathrm{pre}}}(x), f_{\theta_{\mathrm{post}}}(x) \in \mathbb{R}^k$. Given a merged model $\theta_{\mathrm{merged}}$, we define per-sample $\ell_1$ shifts as:

$$\Delta_{\mathrm{pre}}(x) = \left\| f_{\theta_{\mathrm{merged}}}(x) - f_{\theta_{\mathrm{pre}}}(x) \right\|_1, \quad \Delta_{\mathrm{post}}(x) = \left\| f_{\theta_{\mathrm{merged}}}(x) - f_{\theta_{\mathrm{post}}}(x) \right\|_1. \tag{1}$$

Aggregating over the datasets $\mathcal{D}_{\mathrm{pre}}$ and $\mathcal{D}_{\mathrm{post}}$, the total distribution shift is:

$$\Delta_{\mathrm{total}} = \mathbb{E}_{x \sim \mathcal{D}_{\mathrm{pre}}}[\Delta_{\mathrm{pre}}(x)] + \mathbb{E}_{x \sim \mathcal{D}_{\mathrm{post}}}[\Delta_{\mathrm{post}}(x)]. \tag{2}$$

### 3.2 DISTRIBUTION ALIGNMENT VIA SINKHORN DISTANCE

To capture distributional consistency beyond pointwise feature shifts, we leverage entropy-regularized **optimal transport (OT)**. Intuitively, OT seeks the minimal "cost" of transporting the representation distribution of the merged model to match that of task-specific models. This allows us to measure residual misalignment not only at the sample level but also at the distribution level.

Formally, given empirical feature distributions of the pre-task model $\mu_{\mathrm{pre}}$, the post-task model $\mu_{\mathrm{post}}$, and the merged model $\mu_{\mathrm{merged}}$, we compute Sinkhorn distances:

$$\Delta_{\mathrm{pre}}^{\lambda} = \mathcal{L}_{\lambda}(\mu_{\mathrm{pre}}, \mu_{\mathrm{merged}}), \quad \Delta_{\mathrm{post}}^{\lambda} = \mathcal{L}_{\lambda}(\mu_{\mathrm{post}}, \mu_{\mathrm{merged}}),$$

and define the total shift as

$$\Delta_{\mathrm{total}}^{\lambda} = \Delta_{\mathrm{pre}}^{\lambda} + \Delta_{\mathrm{post}}^{\lambda}.$$

Minimizing $\Delta_{\mathrm{total}}^{\lambda}$ ensures that the merged representation remains aligned with both task-specific distributions. This OT-based alignment provides a principled and scalable way to reduce residual distribution shift in continual merging.

**Remark.** Detailed mathematical formulation of Wasserstein distance, Sinkhorn regularization, and computational complexity analysis are provided in the Appendix for completeness.

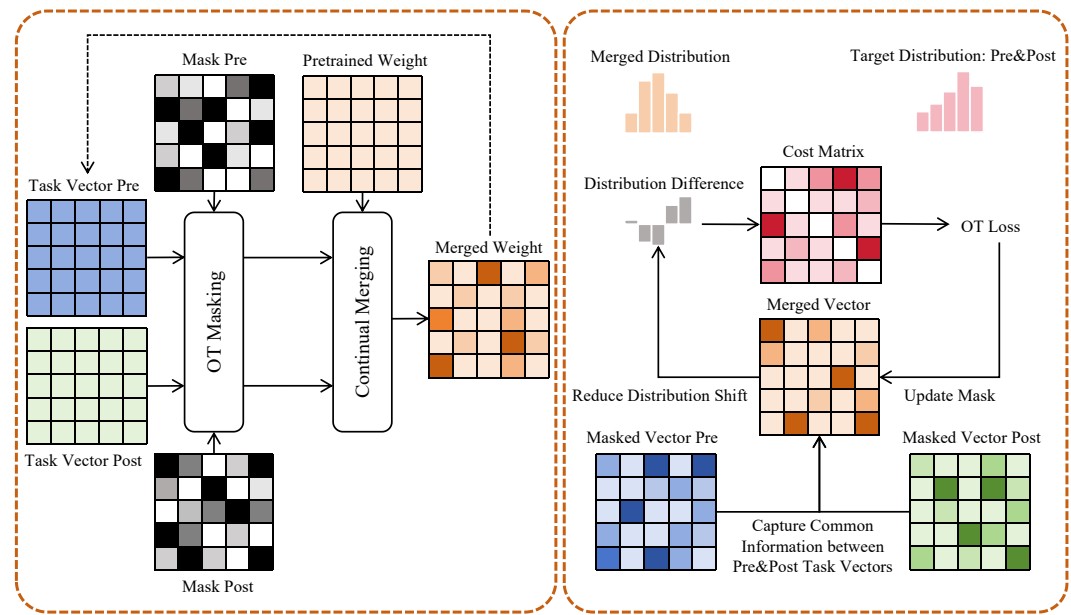

Figure 2: **Left:** Overview of the OTMF continual merging pipeline. Given task vectors from the previous merged model (pre) and the next task's SFT model (post), learnable masks modulate their contributions. The masked vectors are fused and combined with a frozen pretrained model to form the new merged model, which serves as the pre model for the next step, enabling continual task accumulation. **Right:** The OT loss aligns the merged model's features with those of the pre and post models via a cost matrix over feature distributions. This guides mask updates to ensure distributional consistency and knowledge retention.

### 3.3 CONTINUAL FUSION

In the continual setting, task-specific models $\theta^{(1)}, \theta^{(2)}, \ldots, \theta^{(T)}$ arrive sequentially. At each step $t \geq 1$, the goal is to incrementally integrate the new task without accessing earlier data or storing previous models. This imposes two constraints: (i) **constant memory**—only the current merged model and new task model are retained; (ii) **no data replay**—no access to prior training sets. Let $\theta^{(0)}$ be the shared pretrained model. Each task-specific model $\theta^{(k)}$ is represented by its deviation from the backbone as a task vector $\Delta\theta^{(k)} = \theta^{(k)} - \theta^{(0)}$. At step $t$, the current merged task vector $\Delta\theta^{(t-1)}_{\mathrm{m}}$ and the new incoming task vector $\Delta\theta^{(t)}$ are fused via an OT-guided masked fusion operator $\mathcal{F}$, producing an updated merged vector. The final model is then reconstructed by adding this vector back to the backbone:

$$\theta^{(t)} = \theta^{(0)} + \Delta\theta^{(t)}_{\mathrm{m}}, \quad \text{where} \quad \Delta\theta^{(t)}_{\mathrm{m}} = \mathcal{F}(\Delta\theta^{(t-1)}_{\mathrm{m}}, \Delta\theta^{(t)}). \tag{3}$$

The fusion framework $\mathcal{F}$ are optimized to minimize the total Sinkhorn shift $\Delta^\lambda_{\mathrm{total}}$, ensuring that the merged model remains distributionally aligned with previous and incoming tasks.

**Summary:** Our framework combines Sinkhorn-based distribution alignment with continual masked fusion to achieve memory-efficient, previous data-free model integration while mitigating distributional drift across tasks.

## 4 METHODOLOGY

In this section, we introduce OTMF in detail. Subsection 4.1 presents the overall framework and motivation behind our proposed approach, addressing the challenges of catastrophic forgetting and computational efficiency in multi-task learning. Subsection 4.2 elaborates on the fine-grained task vector fusion mechanism enabled by learnable masks. Finally, Subsection 4.3 describes a classification head fine-tuning strategy to mitigate residual distribution shift and further enhance model performance.

## 4.1 OVERALL FRAMEWORK AND MOTIVATION

In multi-task and continual fusion scenarios, parameter merging often leads to catastrophic forgetting due to interference between tasks. Meanwhile, maintaining separate expert models is resource-intensive and scales poorly. To address these challenges, we propose a continual fusion framework built upon a frozen Pretrained Transformer Model (PTM), where task-specific knowledge is encoded as lightweight vector updates and merged through learnable masks guided by optimal transport (OT) plans.

As shown in Figure 2, for each pair of tasks in a continual merging step, we first extract the task-specific vectors—referred to as the "pre" and "post" vectors—which represent the differences between the task-specific models and the frozen PTM. Rather than naively interpolating between these vectors, we introduce a pair of learnable masks that modulate the contribution of each task vector. These masks are applied element-wise to their respective vectors, producing masked updates that are linearly fused into a single merged vector. This merged update is then added to the PTM to form the merged model, which serves as the new base model for future merging.

To be noticed, the merged model obtained at each continual merging step is reused as the "pre-task" model for the next step. This recursive supervision mechanism allows the model to retain knowledge from earlier tasks and incrementally adapt to new ones. By design, this continual accumulation of task-specific information enables the model to maintain performance on prior tasks without storing their full training data or parameter states, thereby achieving efficient continual fusion with minimal forgetting.

## 4.2 TASK VECTOR FUSION VIA LEARNABLE OT MASKS

The core of OTMF framwork lies in the application of learnable masks to the task vectors. These masks, which are initialized uniformly, are optimized to balance the contributions of each task vector during merging. Let $\Delta\theta^{\text{pre}}$ and $\Delta\theta^{\text{post}}$ denote the task vectors, and $M_{\text{pre}}$ and $M_{\text{post}}$ their corresponding masks. We compute the merged task vector as a convex combination of the masked vectors, controlled by a hyperparameter $\alpha$:

$$\Delta\theta_{\text{m}} = \alpha \cdot (M_{\text{pre}} \odot \Delta\theta^{\text{pre}}) + (1-\alpha) \cdot (M_{\text{post}} \odot \Delta\theta^{\text{post}}), \tag{4}$$

where $\odot$ denotes element-wise multiplication. The merged vector is then added to the frozen PTM to construct the merged model. Notably, only the masks are updated during training, ensuring stability in the backbone and task vectors.

A key factor influencing the effectiveness of this fusion is the global hyperparameter $\alpha$. While the learnable masks offer fine-grained control, $\alpha$ dictates the overall preference between retaining past knowledge and incorporating new information. Empirical studies show that setting $\alpha$ around 0.8 yields a desirable trade-off: the merged model effectively learns new tasks while preserving the representational integrity of previous ones. This moderate bias toward prior knowledge plays a crucial role in mitigating catastrophic forgetting during continual merging.

## 4.3 CLASSIFICATION HEAD FINE-TUNING TO MITIGATE RESIDUAL DISTRIBUTION SHIFT

Although the OT-based mask fusion effectively aligns the merged model's distribution with that of task-specific fine-tuned (SFT) models, slight residual misalignment may still remain at the classification head—particularly in ViT architectures, where heads are specialized for their respective output spaces. To further address this issue from a distributional perspective, we adopt a lightweight fine-tuning step after each continual merging stage. Specifically, we update only the current *pre*-task's classification head for 100 epochs using just 25% of the labeled samples, ensuring adaptation to the merged features while strictly respecting the no-replay constraint. The fine-tuning objective is:

$$\min_{W_{\text{cls}}} \mathcal{L}_{\text{cls}}(f_{W_{\text{cls}}}(z_{\text{m}}), y), \tag{5}$$

where $W_{\text{cls}}$ are classifier parameters, $z_{\text{m}}$ the merged features, and $y$ the labels. Despite being deliberately lightweight, this stage consistently enhances generalization by adapting the classification head to the fused representation distribution, thereby complementing the primary benefits of OT-based mask fusion. The ablation of contribution of head fine-tuning( 5 and OT mask are shown at the appendix.

## 4.4 ALGORITHMIC DESCRIPTION

The following pseudo-code summarizes our continual task merging procedure using learnable masks and OT loss:

---
**Algorithm 1:** Continual Task Merging via Learnable Masks and OT Loss

---
**Input:** Frozen pretrained model $\theta^{(0)}$;
Task vectors $\{\Delta\theta^{(1)}, \ldots, \Delta\theta^{(T)}\}$;
Learning rate $\eta$; Trade-off parameter $\alpha$;
Number of OT epochs $E$
**Output:** Final merged task vector $\Delta\theta^{\text{final}}$;
Fine-tuned classification head $W_{\text{cls}}$
$M_{\text{pre}}, M_{\text{post}} \leftarrow \mathbf{1}$ ;                                    // Initialize learnable masks
**for** $t = 2$ **to** $T$ **do**
    **for** $e = 1$ **to** $E$ **do**
        $\Delta\theta_{\text{m}} \leftarrow \alpha \cdot (M_{\text{pre}} \odot \Delta\theta^{(t-1)}) + (1-\alpha) \cdot (M_{\text{post}} \odot \Delta\theta^{(t)})$ ;     // Masked fusion
        $\theta_{\text{m}} \leftarrow \theta^{(0)} + \Delta\theta_{\text{m}}$ ;                                    // Construct merged model
        **if** $e$ *is odd* **then**
            $\mathcal{L}_{\text{OT}} \leftarrow \text{Sinkhorn}(f_{\theta_{\text{m}}}(X), f_{\theta^{(t-1)}}(X))$ ;     // Align with previous task
            $M_{\text{pre}} \leftarrow M_{\text{pre}} - \eta \cdot \nabla_{M_{\text{pre}}}\mathcal{L}_{\text{OT}}$ ;                                    // Update pre-mask
        **else**
            $\mathcal{L}_{\text{OT}} \leftarrow \text{Sinkhorn}(f_{\theta_{\text{m}}}(X), f_{\theta^{(t)}}(X))$ ;           // Align with current task
            $M_{\text{post}} \leftarrow M_{\text{post}} - \eta \cdot \nabla_{M_{\text{post}}}\mathcal{L}_{\text{OT}}$ ;                                    // Update post-mask
        **end**
    **end**
    Fine-tune $W_{\text{cls}}^{(t-1)}$ ;                                    // Optional: update classifier
**end**
**return** $\Delta\theta^{\text{final}} = \Delta\theta_{\text{m}}$

---

**Complexity Analysis.**  Our algorithm introduces only lightweight optimization on learnable masks with the OT loss. For each merging step, the complexity is $O(E \cdot |M|)$, where $E$ is the number of OT epochs (typically $E = 100$ for mask training in our experiments) and $|M|$ denotes the number of mask parameters, which is the same order as the model parameters but updated relatively sparsely. This makes the overall cost substantially lower than training-based merging methods such as *concrete subspace learning*, which typically requires around 1000 epochs of full-parameter optimization.

Compared with training-free approaches (e.g., arithmetic averaging, Ties-Merging), our method indeed incurs additional cost due to OT-based mask updates. However, this small overhead yields significant benefits: in the challenging continual merging scenario, OTMF achieves much higher accuracy and markedly better resistance to forgetting. Moreover, continual merging naturally reduces GPU memory consumption compared to joint merging, since it only loads and processes two task models at a time rather than maintaining all task models simultaneously. Thus, our approach achieves a favorable trade-off, combining the low complexity of merging-style methods with the superior robustness and accuracy often associated with heavy retraining.

## 5 EXPERIMENTS

In this section, we present extensive experiments on both vision and language tasks to comprehensively evaluate the effectiveness of our proposed OTMF framework for continual fusion. We conduct comparisons against various state-of-the-art merging strategies, analyzing both task-specific performance and overall robustness.

### 5.1 EXPERIMENTAL SETUP

For vision tasks, we use pre-trained CLIP-ViT-B/32 and L/14 models evaluated on up to 20 image classification benchmarks, extending prior 8-task joint merging to test scalability. For language tasks, we use Flan-T5-base on GLUE. OT masks are optimized with Adam (lr=0.01, 100 epochs). At each continuous step, only the classification head of the current pre-task model is fine-tuned

Table 1: The performance comparison of continual merging methods. We report the average accuracy and backward transfer (BWT) of the merged models. The best results are highlighted in bold. We abbreviate 'Continual' as 'C.' in the table to save space.

| | Method | ViT-B/32 | | | ViT-L/14 | | |
|---|---|---|---|---|---|---|---|
| | | 8 tasks | 14 tasks | 20 tasks | 8 tasks | 14 tasks | 20 tasks |
| | Pre-Trained | 48.1 | 56.9 | 55.6 | 64.9 | 69.1 | 65.6 |
| | Fine-Tuned | 90.4 | 89.3 | 89.8 | 94.3 | 93.4 | 93.5 |
| | C. Fine-Tuned | 79.8 | 67.4 | 62.6 | 90.0 | 70.9 | 77.7 |
| ACC↑ | Average (SWA) | 66.3±0.0 | 65.4±0.0 | 61.1±0.0 | 80.0±0.0 | 77.5±0.0 | 71.1±0.0 |
| | C. Task Arithmetic | 67.5±0.0 | 66.5±0.0 | 60.6±0.0 | 82.1±0.0 | 77.9±0.0 | 70.3±0.0 |
| | C. Ties-Merging | 49.0±10.2 | 66.2±0.6 | 59.9±0.7 | 64.3±7.0 | 78.0±0.6 | 68.3±0.9 |
| | C. TSVM | 73.9±0.3 | 60.7±0.4 | 58.9±0.3 | 86.6±0.2 | 74.7±0.4 | 74.1±0.5 |
| | OPCM | 75.5±0.5 | 71.9±0.3 | 65.7±0.2 | 87.0±0.4 | **83.5**±0.2 | 76.0±0.2 |
| | **OTMF (Ours)** | **79.7**±0.3 | **75.5**±0.2 | **74.4**±0.3 | **88.0**±0.2 | 83.0±0.5 | **81.2**±0.3 |
| BWT↑ | Average (SWA) | -11.5±2.2 | -8.0±1.3 | -7.1±2.1 | -7.3±1.4 | -5.8±1.0 | -6.4±1.5 |
| | C. Task Arithmetic | -9.6±1.5 | -1.3±0.3 | -3.4±0.4 | -7.1±0.8 | -1.8±0.3 | -3.3±0.3 |
| | C. Ties-Merging | -15.3±8.0 | 1.9±0.6 | -1.5±0.7 | -13.0±5.7 | **1.1**±0.4 | -2.9±1.0 |
| | C. TSVM | -16.1±5.2 | -29.3±4.6 | -28.1±2.7 | -7.1±2.7 | -18.6±3.4 | -16.1±1.5 |
| | OPCM | -6.3±1.1 | -6.0±1.0 | -7.8±1.5 | -2.6±1.0 | -4.3±0.7 | -6.5±1.8 |
| | **OTMF(Ours)** | **3.3**±1.5 | **-4.85**±1.2 | **1.0**±0.5 | **1.9**±0.8 | -1.5±1.4 | **-1.3**±2.1 |

(lr = 0.01, 100 epochs) in 25% of the data, while previously merged heads remain fixed, ensuring localized adaptation without extra data replay. Each merging step requires about 200 lightweight epochs in total (30 s for ViT-B / 32), but produces fine-grained mask fusion that improves resistance to forgetting. Flan-T5 does not require head tuning, so it only requires 100 training steps. Ablations Table 5 (Appendix) shows that $\alpha \approx 0.8$ offers the best trade-off between knowledge preservation and adaptation.

## 5.2 RESULTS AND ANALYSIS

We compare OTMF with representative baselines. For **continual merging**, we include OPCM (sota continual merging methods) (Tang et al., 2025), TSVM (sota joint merging methods) (Gargiulo et al., 2025) and other standard merging methods adapted to the continual setting. For **joint merging**, we consider Weight Averaging, Fisher Merging, Regularized Mean, Task Arithmetic, TIE-Merging, and two subspace merging methods (Tang et al., 2023). Since continual merging is inherently more challenging, we focus on representative comparisons to show that OTMF achieves both strong resistance to forgetting and competitive accuracy. Evaluation includes average and per-task accuracy, with **t-SNE visualizations** illustrating reduced feature shift and **ablation studies** (Appendix) confirming the benefits of OT-based alignment.

### 5.2.1 VISION TASKS

We first examine performance on vision tasks (Tables 1, 2, 3). OTMF consistently outperforms all continual merging baselines, achieving the best average accuracy and superior BWT, thus effectively mitigating catastrophic forgetting. It also surpasses most joint merging methods, with over 5% improvement against the strongest continual baseline (OPCM). While some joint-specific methods are not directly comparable, OTMF's consistent advantage across both continual and joint scenarios highlights its robustness and broad applicability.

### 5.2.2 LANGUAGE TASKS

We conduct similar analyses on language tasks using the Flan-T5-base model. (Appendix)Table 4 reports results across GLUE benchmarks, where OTMF achieves the highest average accuracy of 80.0. Notably, it outperforms all baselines on MNLI, including both continual and joint merging methods, highlighting its effectiveness in addressing distribution shift. By aligning the merged model's feature

Table 2: Multi-task performance of different methods on ViT B-32 across multiple datasets.

| Method | SUN397 | Cars | RESISC45 | EuroSAT | SVHN | GTSRB | MNIST | DTD | Average |
|---|---|---|---|---|---|---|---|---|---|
| Individual | 75.3 | 77.7 | 96.1 | 99.9 | 97.5 | 98.7 | 99.7 | 79.4 | 90.5 |
| Traditional MTL | 73.9 | 74.4 | 93.9 | 98.2 | 95.8 | 98.9 | 99.5 | 77.9 | 88.9 |
| **Joint Merging** | | | | | | | | | |
| Weight Averaging | 65.3 | 63.3 | 71.4 | 73.6 | 64.2 | 52.8 | 87.5 | 50.1 | 66 |
| Fisher Merging | 68.6 | 69.2 | 70.7 | 66.4 | 72.9 | 51.1 | 87.9 | 59.9 | 68.3 |
| RegMean | 65.3 | 63.5 | 75.6 | 78.6 | 78.1 | 67.4 | 93.7 | 52.0 | 71.8 |
| Task Arithmetic | 55.3 | 54.9 | 66.7 | 77.4 | 80.2 | 69.7 | 97.3 | 50.1 | 69 |
| Ties-Merging | 65.0 | 64.3 | 74.7 | 76.8 | 81.3 | 69.4 | 96.5 | 54.3 | 72.8 |
| Concrete TA | 62.5 | 61.1 | 76.0 | 95.7 | 91.0 | 81.9 | 98.5 | 51.9 | 77.3 |
| TW AM | 58.3 | 53.2 | 71.8 | 80.1 | 81.6 | 84.4 | 93.4 | 42.7 | 70.7 |
| TW AM++ | 60.8 | 56.9 | 73.1 | 83.4 | 87.3 | 82.4 | 95.7 | 50.1 | 73.7 |
| TW Concrete AM | 62.7 | 58.9 | 74.5 | 94.8 | 91.1 | 95.0 | 98.1 | 34.6 | 76.2 |
| **Continual Merging** | | | | | | | | | |
| C.TW AM | 50.8 | 51.4 | 54.5 | 52.0 | **77.9** | 78.4 | 93.4 | 71.9 | 66.3 |
| C.LW AM | 62.0 | 59.2 | 60.3 | 43.7 | 48.2 | 86.4 | 67.7 | 20.3 | 56.0 |
| C.Ties-Merging | 61.7 | 59.9 | 60.7 | 40.9 | 35.0 | 33.6 | 61.9 | 57.6 | 51.4 |
| C.Task Arithmetic | **70.7** | 59.6 | 61.2 | 52.1 | 33.5 | 32.2 | 48.0 | 44.6 | 50.2 |
| C.TSVM | 58.8 | 58.6 | 60.9 | 66.1 | 81.7 | **92.5** | 75.9 | **97.3** | 73.4 |
| **OTMF(Ours)** | 68.8 | **66.5** | **77.9** | **87.8** | 77.6 | 89.0 | **97.9** | 71.9 | **79.7** |

Table 3: Multi-task performance of different methods on ViT L-14 across multiple datasets.

| Method | SUN397 | Cars | RESISC45 | EuroSAT | SVHN | GTSRB | MNIST | DTD | Average |
|---|---|---|---|---|---|---|---|---|---|
| Individual | 82.3 | 92.4 | 97.4 | 99.9 | 98.1 | 99.2 | 99.7 | 84.1 | 94.1 |
| Traditional MTL | 80.8 | 90.6 | 96.3 | 96.3 | 97.6 | 99.1 | 99.6 | 84.4 | 93.5 |
| **Joint Merging** | | | | | | | | | |
| Weight Averaging | 72.1 | 81.6 | 82.6 | 91.4 | 78.2 | 70.6 | 97.0 | 62.8 | 79.5 |
| Fisher Merging | 69.2 | 88.6 | 87.5 | 93.5 | 80.6 | 74.8 | 93.3 | 70.0 | 82.2 |
| RegMean | 73.3 | 81.8 | 86.1 | 97.0 | 88.0 | 84.2 | 98.5 | 60.8 | 83.7 |
| Task Arithmetic | 82.1 | 65.6 | 92.6 | 86.8 | 98.9 | 86.7 | 74.1 | 87.9 | 84.4 |
| Ties Merging | 84.5 | 67.7 | 94.3 | 82.1 | 98.7 | 88.0 | 75.0 | 85.7 | 84.5 |
| Concrete TA | 86.2 | 66.9 | 96.7 | 93.4 | 99.1 | 89.0 | 74.6 | 93.6 | 87.4 |
| **Continual Merging** | | | | | | | | | |
| C.TW AM | 70.2 | 77.7 | 79.5 | 80.0 | 85.1 | 94.3 | 97.8 | 82.1 | 83.3 |
| C.LW AM | 68.5 | 77.8 | 71.9 | 66.4 | 68.3 | 60.4 | 96.3 | 75.2 | 73.1 |
| C.Ties-Merging | 74.2 | 78.2 | 71.6 | 61.0 | 58.2 | 51.0 | 75.5 | 55.0 | 65.6 |
| C.Task Arithmetic | **76.5** | 77.9 | 71.2 | 61.0 | 58.0 | 50.8 | 74.8 | 55.1 | 65.7 |
| C.TSVM | **76.5** | 77.9 | 71.2 | 61.0 | 58.0 | 50.8 | 74.8 | 55.1 | 65.7 |
| **OTMF(Ours)** | 76.3 | **84.5** | **87.9** | **93.9** | **87.3** | **94.6** | **99.3** | **79.9** | **88.0** |

distribution with that of the SFT model, OTMF improves performance while mitigating catastrophic forgetting.

### 5.2.3 DISTRIBUTION VISUALIZATION

To further assess the effectiveness of OTMF, we conduct a qualitative visualization study to analyze its impact on the feature distribution of merged models. Specifically, we apply t-SNE to visualize the output distributions of the merged models along with their corresponding each continual merging step's task pre models (The final step will include both the pre and post models). The visualizations reveal that OTMF achieves a significantly finer alignment between the merged model's distribution and those of task pre models. Compared to other continual fusion strategies, OTMF yields the smallest distribution shift, preserving the structural semantics of both task pre and post SFT models. This indicates that the OT mask serves as an effective tool to reconcile task-specific knowledge during fusion, leading to more stable and reliable multi-task performance. In the main text, we only present the t-SNE comparison between OTMF and Task-wise AdaMerging, while additional t-SNE plots 6, 7, 8, 9, 10, for other methods are provided in the appendix, where the distribution shifts during fusion and the effectiveness of OTMF in alleviating such shifts can be more intuitively observed.

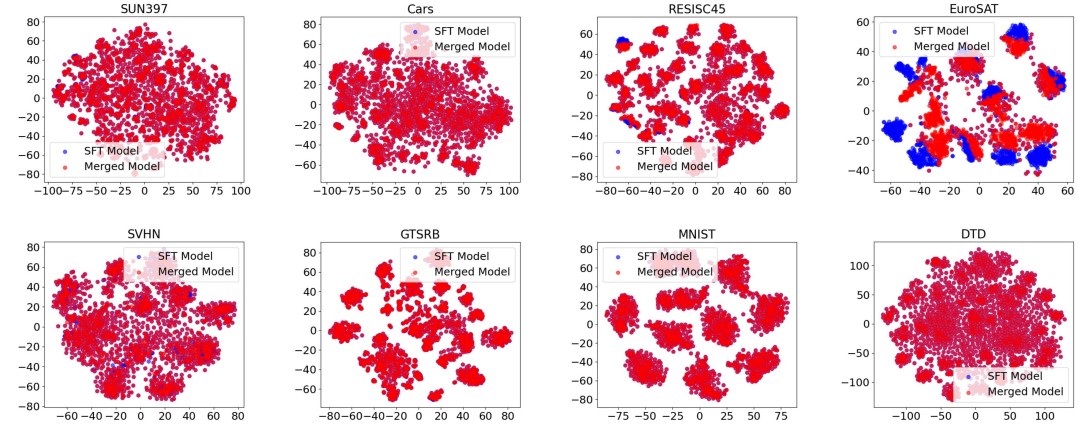

Figure 3: The Distribution Visualization of Continual OT Mask Merging

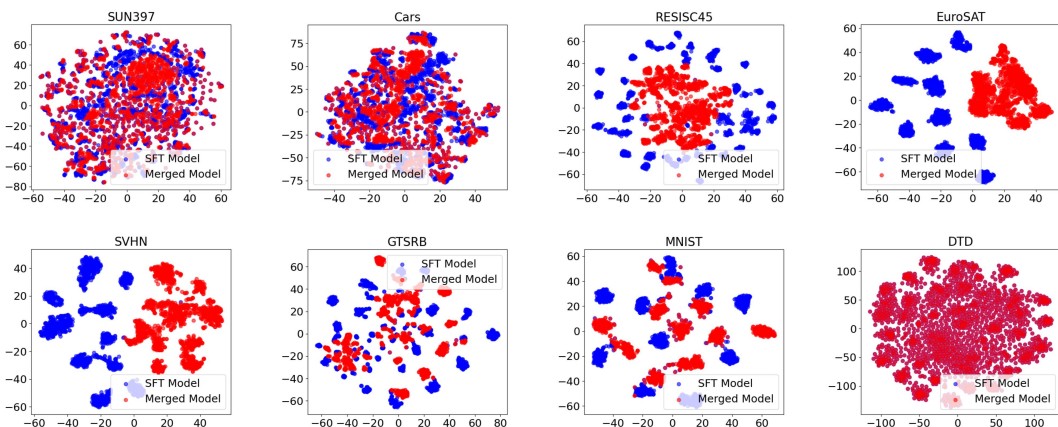

Figure 4: The Distribution Visualization of Continual Task-wise Adamerging

# 6 CONCLUSION

We propose OTMF, a model fusion framework based on Optimal Transport to mitigate distribution shift in multi-task and continual fusion. By identifying task-agnostic masks that minimize OT-based divergence between merged and task-specific distributions, OTMF preserves semantic structure and reduces parameter interference. Experiments on vision and language benchmarks show that OTMF outperforms prior joint and continual merging methods, achieving higher accuracy and improved memory efficiency. The framework supports scalable, replay-free integration of tasks, enabling continual fusion without catastrophic forgetting.

Despite these strengths, OTMF still presents several limitations. First, it assumes that the incoming task vectors are reasonably aligned in distribution space, which may not hold in scenarios involving highly heterogeneous domains or drastically dissimilar model fine-tuning objectives. In such cases, the learned optimal transport plan may converge to suboptimal mappings, limiting the model's ability to extract transferable components. Second, the use of learnable masks, though effective for capturing fine-grained common information, necessitates extra training and introduces additional memory overhead from storing mask parameters. Lastly, for ViT models, classification head fine-tuning introduces a small amount of additional training overhead, serving as an auxiliary step to further align residual distribution shift after merging.

Nonetheless, OTMF provides a principled and effective solution for continual model merging in scenarios where task-specific data or simultaneous model access is restricted. We hope this work inspires further research into transport-based distribution alignment, especially under non-ideal, heterogeneous, and real-world deployment conditions.

ETHICS STATEMENT

All experiments in this paper fuse publicly released AI models and rely exclusively on open datasets and pre-trained checkpoints. No human or animal subjects, proprietary data, or sensitive information are involved; consequently, no ethical concerns arise.

REPRODUCIBILITY STATEMENT

Our complete codebase is bundled in the supplementary ZIP file. Reviewers can unzip it, install the listed dependencies, and reproduce all results by downloading the already-public datasets and model checkpoints cited in the paper. Upon acceptance, the repository will also be released on GitHub.

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

# A APPENDIX

## A.1 LANGUAGE TASK EXPERIMENTS

Table 4: Multi-task performance of different methods on Flan-T5-base across multiple datasets.

| Method | CoLA | MNLI | MRPC | QNLI | QQP | RTE | SST2 | STSB | Avg |
|---|---|---|---|---|---|---|---|---|---|
| Individual | 69.1 | 82.7 | 85.5 | 90.9 | 84.0 | 84.4 | 92.9 | 87.4 | 84.6 |
| **Joint Merging** | | | | | | | | | |
| Weight Averaging | 69.7 | 59.7 | 78.9 | 90.1 | 83.8 | 80.5 | 91.2 | 72.0 | 78.2 |
| Task Arithmetic | 68.8 | 55.2 | 78.7 | 89.8 | 83.7 | 79.1 | 91.5 | 72.4 | 77.4 |
| Ties-Merging | 68.3 | 56.3 | 79.4 | 89.8 | 83.7 | 79.4 | 91.6 | 71.2 | 77.5 |
| Concrete TA | 69.1 | 58.1 | 78.4 | 89.9 | 83.5 | 79.4 | 91.6 | 73.4 | 78.0 |
| LW AM | 69.1 | 60.3 | 78.4 | 90.0 | 83.6 | 79.1 | 91.6 | 74.1 | 78.3 |
| LW Concrete AM | 69.0 | 59.4 | 80.1 | 89.9 | 82.9 | 79.1 | 91.7 | 75.4 | 78.5 |
| **Continual Merging** | | | | | | | | | |
| C.TW AM | 68.6 | 58.2 | 77.5 | **90.0** | **83.9** | **81.9** | 92.5 | 73.2 | 78.2 |
| C.LW AM | 69.9 | 64.5 | **80.4** | 89.8 | 81.9 | 78.0 | **93.1** | **79.2** | 79.6 |
| C.Ties-Merging | 69.1 | 56.5 | 76.2 | 88.4 | 82.1 | 80.1 | 91.2 | 62.2 | 75.7 |
| C.Task Arithmetic | **70.5** | 57.8 | 78.4 | **90.2** | 83.6 | 80.5 | 92.3 | 77.8 | 78.9 |
| **OTMF(Ours)** | 69.8 | **77.3** | 78.7 | 87.3 | 82.6 | 78.7 | 92.0 | 73.4 | **80.0** |

## A.2 ABLATION STUDIES

**Merging Hyper Parameter** $\alpha$  We conduct a thorough fusion study on the merging coefficient $\alpha$. As shown in Table 5, setting $\alpha 0.8$ consistently yields the best trade-off: the OT-MF variant absorbs the maximum amount of common information while preserving the original expert knowledge (i.e., strongest anti-forgetting ability). Smaller values under-merge, whereas larger values begin to overwrite specialist weights, confirming that $\alpha=0.8$ is a robust sweet spot for OT-MF.

Table 5: Ablation study of parameter $\alpha$ across multiple datasets.

| $\alpha$ | SUN397 | Cars | RESISC45 | EuroSAT | SVHN | GTSRB | MNIST | DTD | Avg |
|---|---|---|---|---|---|---|---|---|---|
| 0 | 0.516 | 0.514 | 0.381 | 0.444 | 0.340 | 0.404 | 0.856 | **0.981** | 0.554 |
| 0.1 | 0.496 | 0.488 | 0.385 | 0.424 | 0.366 | 0.370 | 0.775 | 0.976 | 0.535 |
| 0.2 | 0.512 | 0.516 | 0.426 | 0.439 | 0.366 | 0.426 | 0.767 | 0.978 | 0.554 |
| 0.3 | 0.537 | 0.514 | 0.474 | 0.387 | 0.373 | 0.546 | 0.776 | 0.978 | 0.573 |
| 0.4 | 0.561 | 0.543 | 0.534 | 0.467 | 0.532 | 0.673 | 0.824 | 0.975 | 0.639 |
| 0.5 | 0.575 | 0.554 | 0.577 | 0.526 | 0.581 | 0.780 | 0.842 | 0.974 | 0.676 |
| 0.6 | 0.597 | 0.583 | 0.622 | 0.637 | 0.687 | 0.861 | 0.840 | 0.932 | 0.720 |
| 0.7 | 0.623 | 0.581 | 0.694 | 0.773 | 0.731 | 0.860 | 0.867 | 0.869 | 0.764 |
| **0.8** | 0.689 | **0.667** | **0.779** | **0.873** | **0.789** | **0.889** | **0.978** | 0.717 | **0.797** |
| 0.9 | 0.728 | 0.599 | 0.728 | 0.805 | 0.595 | 0.708 | 0.826 | 0.639 | 0.710 |
| 1 | **0.762** | 0.599 | 0.691 | 0.746 | 0.414 | 0.636 | 0.866 | 0.552 | 0.658 |

**OT Mask and Head finetuning**  To better understand the contribution of each component in our OTMF framework, we conduct two ablation studies: one without classification head tuning and the other without OT mask training. In the first setting, we fix the classification heads during merging, which leads to a performance drop—from **79.7** to **72.2** on ViT-B/32 and from **88.0** to **82.7** on ViT-L/14—as shown in Figure 5. This demonstrates that classification head tuning facilitates adaptation to the residual distribution shift in the output space, and further underscores the foundational role of OT mask training in aligning output distributions and eliminating such shift. In contrast, when we disable OT mask learning and instead use task-vectors directly for merging, performance degrades substantially: accuracy on ViT-B/32 drops from **79.7** to **64.0**, ViT-L/14 from **88.0** to **64.0**, and Flan-T5-base from **79.5** to **78.3**. These results highlight the critical role of OT mask learning in

achieving distributional alignment and effective task integration. Overall, the ablation studies confirm that OT mask learning is the primary driver of performance gains, while classifier head fine-tuning provides complementary improvements.

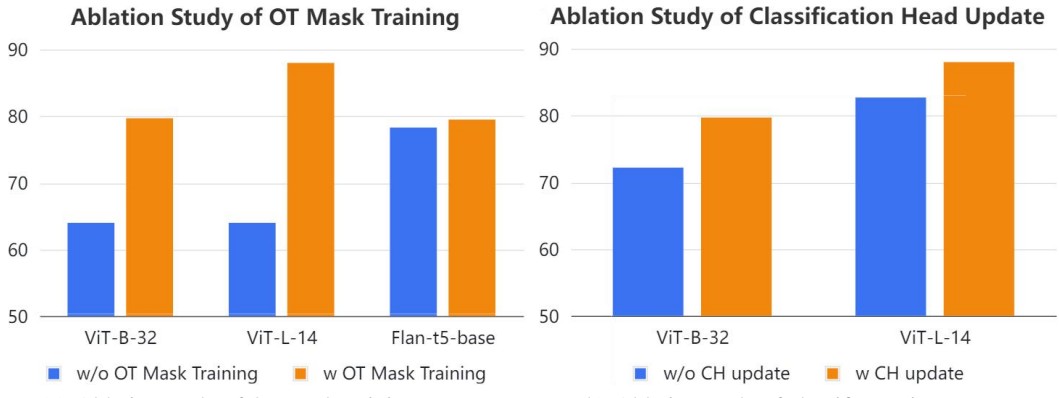

(a) Ablation study of OT mask training, avgacc     (b) Ablation study of classifier tuning, avgacc

Figure 5: Ablation Study of OTMF

### A.3 T-SNE Distribution of Different Merging Methods

To further illustrate the distributional effects of different continual fusion methods, we provide additional t-SNE visualizations in Figure 6, 7, 8, 9, 10. These figures complement the main text, where only OTMF and Task-wise AdaMerging were compared. By including more methods, readers can more intuitively observe the distribution shifts occurring during the fusion process, as well as the superior ability of OTMF to mitigate such shifts and preserve task-specific semantics.

### A.4 Sinkhorn Algorithm, Formulation and Complexity

**Algorithm and Formulation.** To address the high computational cost of classical Optimal Transport (OT), we employ the **Sinkhorn algorithm** Cuturi (2013), which introduces entropic regularization to make OT tractable on large-scale and high-dimensional data. Instead of solving the original OT linear program:

$$d_M(r,c) = \min_{P \in U(r,c)} \langle P, M \rangle,$$

where $U(r,c)$ denotes the transportation polytope with fixed marginals $r, c \in \Sigma_d$, and $M \in \mathbb{R}^{d \times d}_+$ is the ground cost matrix, Sinkhorn regularization adds a negative entropy term:

$$d_M^\lambda(r,c) = \min_{P \in U(r,c)} \langle P, M \rangle - \frac{1}{\lambda} h(P),$$

where $h(P) = -\sum_{i,j} p_{ij} \log p_{ij}$ is the entropy of the coupling matrix $P$, and $\lambda > 0$ is a regularization strength.

This strictly convex formulation has a closed-form solution in the form:

$$P^\lambda = \mathrm{diag}(u) \cdot K \cdot \mathrm{diag}(v), \quad \text{with} \quad K = \exp(-\lambda M),$$

where vectors $u$ and $v$ are computed efficiently using the Sinkhorn-Knopp fixed-point updates to match the marginal constraints:

$$u^{(t+1)} = \frac{r}{Kv^{(t)}}, \quad v^{(t+1)} = \frac{c}{K^T u^{(t+1)}}.$$

Compared to the classic OT solution which lies at the vertices of $U(r,c)$, the Sinkhorn solution yields a smoother transport plan with higher entropy. This result shows how the optimal plan lies within a KL-divergence ball around the independence table $rc^T$, effectively balancing transport cost and plan entropy. The entropic constraint leads to faster convergence and robustness, particularly valuable in high-dimensional learning applications.

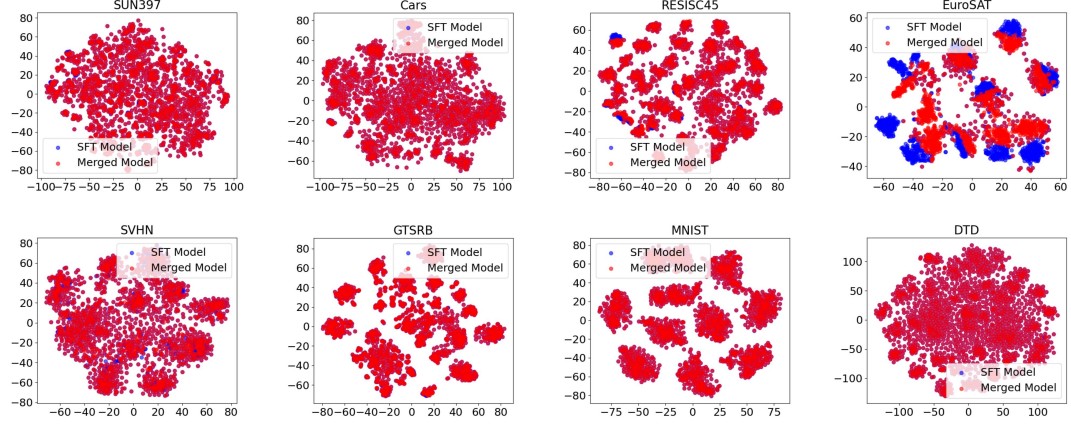

Figure 6: The Distribution Visualization of OTMF Merging

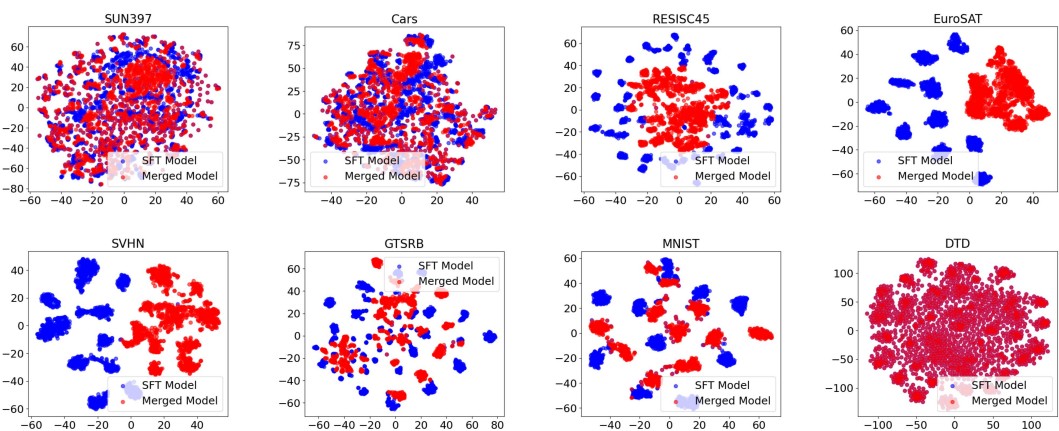

Figure 7: The Distribution Visualization of Continual Taskwise Adamerging

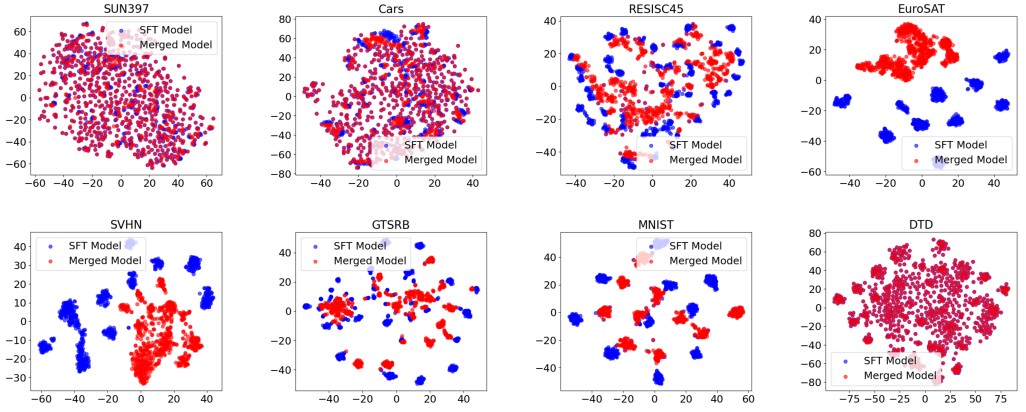

Figure 8: The Distribution Visualization of Continual Layerwise Adamerging

**Complexity.** The Sinkhorn algorithm has a computational complexity of $\mathcal{O}(n^2 \log(1/\varepsilon))$ for achieving an $\varepsilon$-accurate solution, where $n$ is the number of samples in the empirical distributions. This is significantly more efficient than the $\mathcal{O}(n^3)$ complexity of exact Wasserstein distance computation. In our setting, with $n_{\text{pre}}$ and $n_{\text{post}}$ samples from the pre- and post-task datasets, the total complexity for computing $\Delta_{\text{pre}}^{\lambda}$ and $\Delta_{\text{post}}^{\lambda}$ is $\mathcal{O}((n_{\text{pre}}^2 + n_{\text{post}}^2) \log(1/\varepsilon))$. Given that $n_{\text{pre}}, n_{\text{post}} \ll N$ (where $N$ is

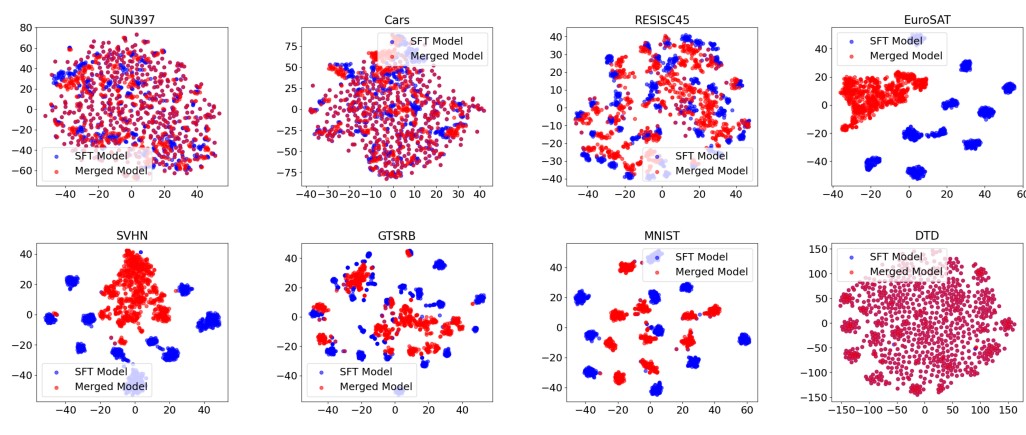

Figure 9: The Distribution Visualization of Continual Task Arithmetic Merging

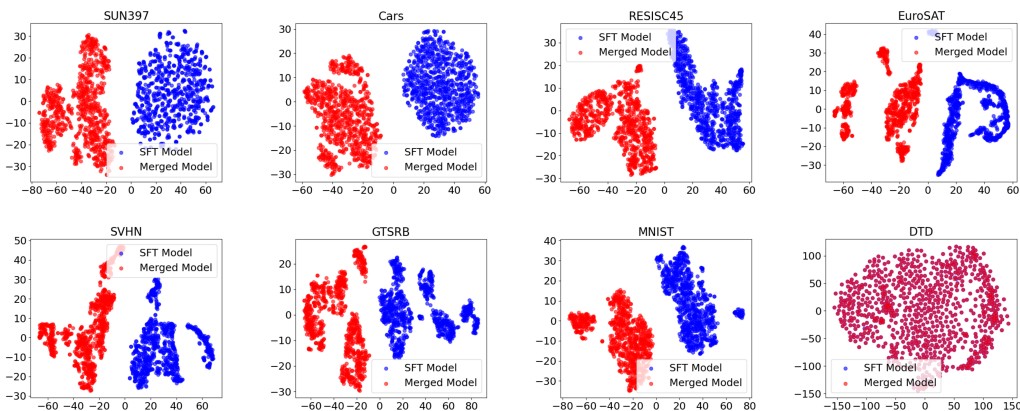

Figure 10: The Distribution Visualization of Continual TIE Merging

the total training set size in typical continual learning settings), this approach remains computationally feasible even for large-scale datasets.

### A.5 POT LIBRARY FOR SINKHORN LOSS COMPUTATION

In our implementation of the Sinkhorn loss to mitigate distribution shift, we utilize the **POT** (Python Optimal Transport) library Flamary et al. (2021), a dedicated Python toolbox for solving optimal transport (OT) problems in machine learning.

The POT library provides a highly efficient and numerically stable implementation of the entropic-regularized OT solver via the Sinkhorn-Knopp algorithm. Specifically, we used the `ot.sinkhorn` function, which minimizes the regularized transport cost between two discrete probability distributions $a$ and $b$ under a ground cost matrix $M$. The formulation is:

$$\text{Sinkhorn}_\varepsilon(a, b) = \min_{T \in \Pi(a,b)} \langle T, M \rangle - \varepsilon H(T), \tag{6}$$

where $\Pi(a, b)$ denotes the set of doubly stochastic matrices satisfying marginal constraints, and $H(T) = -\sum_{i,j} T_{i,j}(\log T_{i,j} - 1)$ is the entropy regularizer. The entropy coefficient $\varepsilon > 0$ controls the smoothness of the optimal transport plan $T$. This formulation enables differentiable and fast OT computations, making it suitable for integration into modern machine learning pipelines.

POT is modular and includes sub-packages such as `ot.lp` for exact linear programming solvers, `ot.gromov` for Gromov-Wasserstein problems, and `ot.da` for domain adaptation with OT. Its syntax is user-friendly and consistent, facilitating fast prototyping of OT-based loss functions. In

our case, it allowed for efficient and stable computation of Sinkhorn divergence in the training loss without requiring manual implementation of the optimization routine.

The library is open-source (MIT license), well-documented, and actively maintained, with over 20 contributors and extensive unit tests. It depends only on standard Python scientific computing libraries such as NumPy and SciPy.

For further technical details, please refer to the official publication and documentation:

- R. Flamary et al., "POT: Python Optimal Transport," *Journal of Machine Learning Research*, vol. 22, pp. 1–8, 2021. `http://jmlr.org/papers/v22/20-451.html`
- GitHub repository: `https://github.com/PythonOT/POT`

## A.6 DATASETS AND PREPROCESSING

**Vision Datasets.** In our experiments on vision tasks, we adopt the pre-trained CLIP models—`ViT-B/32` and `ViT-L/14`—as the backbone encoders. The continual fusion process is evaluated on eight widely-used image classification benchmarks, covering diverse visual domains:

- **SUN397**: A large-scale scene recognition dataset with 397 categories covering diverse environments, including indoor, outdoor, man-made, and natural scenes. It provides a comprehensive benchmark for evaluating models' capability in scene understanding.

- **Stanford Cars**: A fine-grained visual categorization dataset containing 16,185 images across 196 car classes, defined by make, model, and year. It is widely used to benchmark fine-grained recognition performance.

- **RESISC45**: A remote sensing image dataset with 45 scene classes, each containing 700 images. It represents a variety of land-use and land-cover patterns captured from aerial and satellite imagery, useful for remote sensing classification tasks.

- **EuroSAT**: A satellite image dataset with 10 classes derived from Sentinel-2 imagery. It covers land use and land cover categories such as residential areas, forests, rivers, and agricultural fields.

- **SVHN**: The Street View House Numbers dataset consisting of over 600,000 digit images cropped from real-world house numbers in Google Street View. It is a challenging benchmark for digit recognition under natural image conditions.

- **GTSRB**: The German Traffic Sign Recognition Benchmark containing more than 50,000 images of 43 traffic sign categories. It evaluates models' ability to handle real-world variations in illumination, occlusion, and resolution.

- **MNIST**: A classic benchmark dataset of handwritten digits (0–9), containing 70,000 grayscale images. It is widely used as an entry-level testbed for evaluating image classification methods.

- **DTD**: The Describable Textures Dataset, comprising 47 texture categories defined by semantic attributes such as "striped," "dotted," or "zigzagged." It provides a benchmark for texture recognition and attribute-based learning.

- **Flowers102**: A fine-grained classification dataset with 102 flower categories. Each class contains between 40 and 258 images, covering a wide variety of flower species with significant intra-class variability.

- **PCAM**: The PatchCamelyon dataset, derived from the Camelyon16 challenge, contains histopathology image patches (binary labels: tumor vs. normal tissue). It serves as a medical imaging benchmark for cancer metastasis detection.

- **FER2013**: A facial expression recognition dataset with 35,887 grayscale images of human faces annotated with seven emotion categories (e.g., happy, sad, angry). It is widely used in affective computing and emotion recognition research.

- **Oxford-IIIT Pet**: A dataset containing 37 categories of pet images (cats and dogs), with roughly 200 images per class. Each image includes class labels and pixel-level segmentation annotations.

- **STL-10**: A dataset designed for developing semi-supervised learning methods, with 10 object classes and 100,000 unlabeled images in addition to 13,000 labeled images. The images have higher resolution (96×96) than CIFAR datasets.

- **CIFAR-100**: A dataset of 60,000 images across 100 object classes, with 600 images per class. It is considered a challenging benchmark for image classification due to its fine-grained categories and small image size (32×32).

- **CIFAR-10**: A subset of CIFAR with 10 object classes, including airplanes, birds, and trucks. It contains 60,000 images (50,000 training and 10,000 testing) and is one of the most widely used benchmarks for image recognition.

- **Food-101**: A large-scale dataset with 101 categories of food, containing 101,000 images in total. Each class has 1,000 images, making it a standard benchmark for food recognition tasks.

- **Fashion-MNIST**: A dataset of Zalando's article images, comprising 70,000 grayscale images across 10 categories of clothing (e.g., shirts, shoes, bags). It is considered a more challenging drop-in replacement for MNIST.

- **EMNIST**: The Extended MNIST dataset containing handwritten character digits and letters. It includes multiple splits (e.g., ByClass, ByMerge, Balanced) covering up to 62 classes (digits and uppercase/lowercase letters).

- **KMNIST**: The Kuzushiji-MNIST dataset, a replacement for MNIST with 70,000 grayscale images of Japanese Kuzushiji characters from 10 classes. It serves as a benchmark for non-Latin handwritten character recognition.

- **Rendered SST-2**: A visual sentiment classification dataset derived from the Stanford Sentiment Treebank (SST-2). Sentiment labels (positive/negative) are rendered into images, enabling multimodal evaluation of sentiment analysis.

These datasets were selected to reflect various visual challenges, from coarse scene understanding to fine-grained recognition and texture classification. During continual merging, the classification head of the previous model is fine-tuned using a small number of labeled samples per task (learning rate = 0.001, 1000 epochs), while the backbone parameters remain frozen. This ensures alignment between the new merged feature space and the decision boundaries.

**Language Datasets.** For language tasks, we utilize the pre-trained `Flan-T5-base` model. Evaluation is conducted on eight GLUE benchmark tasks, each addressing distinct natural language understanding capabilities:

- **CoLA**: Corpus of Linguistic Acceptability for syntactic judgment classification.
- **MNLI**: Multi-Genre Natural Language Inference for recognizing textual entailment.
- **MRPC**: Microsoft Research Paraphrase Corpus for paraphrase detection.
- **QNLI**: Question Natural Language Inference, a QA-style NLI dataset.
- **QQP**: Quora Question Pairs for detecting duplicate questions.
- **RTE**: Recognizing Textual Entailment binary classification task.
- **SST-2**: Stanford Sentiment Treebank for binary sentiment classification.
- **STS-B**: Semantic Textual Similarity Benchmark for regression of sentence similarity.

Unlike vision tasks, the Flan-T5 models do not involve classification heads; thus, only OT mask-based merging is applied in the language setting. The datasets were selected to ensure a broad coverage of syntactic, semantic, and inference-related language phenomena.

**Preprocessing and Setup.** For all datasets, standard splits are used. Features are extracted from the final encoder layer of each backbone (CLIP or Flan-T5), and normalized before computing OT-based distances. The optimal transport masks are optimized using the Adam optimizer (learning rate = 0.01, 100 epochs). The fusion hyperparameter $\alpha$ is empirically set to 0.8 across all tasks, balancing retention of previous knowledge and adaptation to new information.

This design enables fair, scalable, and memory-efficient evaluation of continual model fusion across both vision and language modalities.

## A.7 BASELINES

The baseline methods considered in our experiments are divided into two categories: (i) non-model merging methods and (ii) model merging methods. Among the model merging baselines, a selected subset is adapted to the continual fusion setting to ensure a fair comparison with our proposed OTMF framework.

(i) *Non-model merging methods*:

- **Pretrained**: Evaluates the frozen pre-trained model directly on downstream tasks without any task-specific adaptation. This method incurs minimal computational cost but exhibits weak performance due to the lack of task alignment.

- **Individual**: Independently fine-tunes a separate model for each task. While this yields strong accuracy due to task isolation, it is memory-inefficient as it stores multiple models.

- **Traditional MTL** Zhang et al. (2022): Trains a shared-parameter model jointly on all tasks using multi-task supervision. It is compact but susceptible to negative transfer when task conflicts arise.

(ii) *Model merging methods*:

- **Weight Averaging**: Directly averages the weights of models trained on different tasks. Simple but assumes linear connectivity in parameter space.

- **Fisher Merging** Matena & Raffel (2022a): Weights parameters using their estimated importance from the Fisher information matrix Fisher (1922), giving more influence to sensitive parameters.

- **RegMean** Jin et al. (2023): Merges models via weighted averaging based on second-order statistics derived from training data.

- **Task Arithmetic** Ilharco et al. (2023): Constructs task vectors as parameter deltas relative to a pre-trained model and merges them additively to form a fused representation.

- **Ties-Merging** Yadav et al. (2023): Improves on Task Arithmetic by trimming noisy parameters and aligning signs before merging.

- **Concrete TA** Tang et al. (2023): Performs model fusion in a masked subspace of parameters using learnable binary masks, incorporating Task Arithmetic.

- **Concrete AM** Tang et al. (2023): Enhances Concrete TA by using AdaMerging **?** to learn adaptive coefficients for each task.

- **TW AdaMerging** Yang et al. (2024d): Learns task-wise merging weights from an unlabeled validation set to adaptively combine task vectors.

- **LW AdaMerging** Yang et al. (2024d): A layer-wise version of TW AdaMerging that estimates merging weights for each layer separately.

**Continual Merging Baselines.** To assess continual fusion performance, we adapt several representative model merging methods into the continual setting. Specifically, the following methods are included in our continual merging evaluation:

- **Task Arithmetic**: Task vectors are sequentially added to the merged model across tasks without re-accessing prior models.

- **Ties-Merging**: Extends Task Arithmetic with parameter filtering and sign correction before iterative fusion.

- **TW AdaMerging**: Estimates task-level merging coefficients for each fusion step using an unlabeled test set.

- **LW AdaMerging**: A fine-grained version of TW AdaMerging, estimating per-layer weights for merging.

- **TSVM**(Gargiulo et al., 2025): A SOTA joint merging method using low rank decompose to merge task-vectors without conflict.

- **OPCM**(Tang et al., 2025): A SOTA continual merging method using optimal parameter consolidation for stable knowledge integration.

All continual baselines follow the same fusion protocol as our method. At each continual merging step $t$, the previously merged model and the incoming task-specific model are merged using their respective training datasets. Only current and incoming task data are accessible; no prior data replay or model storage is assumed. This setting ensures fair comparison under strict continual fusion constraints and isolates the effect of each merging strategy.

## A.8 IMPACT STATEMENT

This work presents a scalable and memory-efficient framework for continual model fusion via optimal transport-based mask learning. By aligning task representations in latent space, our method mitigates distribution shift and prevents forgetting without accessing previous models or data. This has practical value for real-world applications such as federated learning, edge computing, and privacy-sensitive domains, where task-wise model integration is required without full retraining. As our approach operates on public benchmarks without relying on sensitive or personalized information, it raises minimal ethical concern. Nonetheless, future extensions should consider safeguards to prevent misuse in unintended model compositions.

## A.9 DEVICES

All experiments, including both the optimal transport mask training and the continual model fusion procedures, are conducted on a dedicated Linux server equipped with high-performance computing hardware. The server hosts two Intel(R) Xeon(R) Silver 4210 CPUs running at 2.20GHz, comprising 20 physical cores (40 logical threads) in total. For GPU acceleration, we utilize NVIDIA Tesla V100-SXM2 GPUs, each with 32GB of memory, supported by CUDA 12.2 and driver version 535.216.01. All models and baselines are implemented using the PyTorch framework. This setup ensures both memory and compute scalability for efficient model merging and evaluation across large-scale multi-task benchmarks.

## A.10 USE OF LARGE LANGUAGE MODELS (LLMS)

We utilized Large Language Models (LLMs) solely for grammar polishing and language refinement.

