# OpenReview forum: "Merging without Forgetting: Continual Fusion of Task-Specific Models via Optimal Transport"
_ICLR.cc/2026/Conference — Submitted to ICLR 2026_

### Official Review · Reviewer_M6mR · 2025-10-22

**Soundness:** 3
**Presentation:** 2
**Contribution:** 2
**Rating:** 4
**Confidence:** 4

**Summary:**

The paper proposes OTMF, a method for continual model merging using optimal transport to align feature distributions and learn masks on task vectors. It aims to mitigate distribution shifts caused by naive parameter interpolation, enabling continual fusion with bounded memory.

**Strengths:**

1. The core idea of using optimal transport to guide learnable masks for aligning pre- and post-task distributions is conceptually interesting and addresses a real limitation in parameter-space merging methods (e.g., catastrophic forgetting due to distribution shifts, especially under continual model merging setting where previous task-specific models are not available).
2. The continual fusion paradigm, which only requires the current merged model and the new task model, is practical for scalability and memory efficiency, distinguishing it from joint merging approaches that need all models simultaneously.

**Weaknesses:**

1. A significant concern arises from OTMF's reliance on **labeled data** for fine-tuning the classification heads after each merging step. The using of labeled data is atypical in the model merging literature, techniques like task arithmetic, ties merging and opcm are data-free methods, and AdaMerging only using unlabeled data for test-time adaptation.
2. The performance of merged model without classification heads tuning should also be listed at Table 1.
3. Only high-level results such as average accuracy are shown in the tables, the authors could provide some more detailed results (e.g. per-task accuracy during the continual model merging at each merging step).
4. Insufficient Detail on the Optimization Objective in the Main Text. The detail about optimization objective ($\mathcal{L}_{\lambda}$) should be introduced in the main text.

**Questions:**

1. Provide full experimental tables: inclduing per-task accuracies, forgetting curves.
2. How does OTMF handle task similarity? E.g., if task vectors are orthogonal, does masking suffice?

---

> ### Comment · Reviewer_M6mR · 2025-11-27
>
> I quite like the idea of identifying a shared mask. To strengthen the manuscript, the presentation and experiments could be further refined.

---

### Official Review · Reviewer_SLrx · 2025-10-26

**Soundness:** 2
**Presentation:** 2
**Contribution:** 2
**Rating:** 2
**Confidence:** 4

**Summary:**

The paper tackles the problem of continual model merging, which consists of sequentially merging models as they appear over time. The authors propose Optimal Transport-based Masked Fusion (OTMF) to address this problem. During each new task, the method learns two masks — one associated with the current task vector and the other with the previous one. At the end of the task, these two masks are applied to their respective task vectors, and the resulting masked vectors are combined through a weighted sum with an empirically chosen factor $\alpha$. The resulting merged task vector represents the updated merged task vector, which then becomes the “previous” task vector for the next iteration.
The training of the masks is performed in an unsupervised manner using the Sinkhorn loss, from the optimal transport loss theory which aims to mitigate the distribution shift between consecutive task vectors. After each task, the classifier head is fine-tuned using 25% of the labeled data from the current task to adapt to the fused representation.
The proposed method is evaluated on both vision tasks  and language tasks.

**Strengths:**

-  The use of the Sinkhorn distance in the context of continual model merging is novel.
- Extensive comparisons on both vision and language tasks demonstrate that the proposed method achieves good performance.

**Weaknesses:**

1) The paper is poorly written in terms of structure and contains numerous grammatical and spelling errors. In the following paragraphs more details:

- The narrative of the paper is difficult to follow and makes it difficult to understand how the proposed methodology works.  Section 3.3 (within the Preliminaries) already describes part of the overall framework and should therefore be merged with Section 4.1. In its current form, the presentation is fragmented: the Sinkhorn loss from optimal transport theory—one of the main contributions—does not appear in the Methodology section but in preliminaries. The entire Methodology should be reorganized to flow more coherently: it should first introduce the overall framework, then clearly explain how the Sinkhorn loss operates (this explanation should not be relegated to the appendix, since it constitutes a core contribution), and finally describe how this loss is used to train the learnable masks and perform continual fusion with the classifier training.

- The figures need significant improvement. The left part of Figure 1 does not illustrate any aspect of optimal transport theory—it is abstract, contains histograms whose represented distributions are unclear, and fails to provide meaningful insight into how the proposed method works. The right part of Figure 1 could instead serve as a small teaser in the introduction. A new Figure 1 should be created to clearly explain the overall method and its key components and should be merged with Figure 2 (that as well needs improvements).
- For me, only the algorithm makes it clear how the method works, but  a short description of the algorithm  is missing.
- There are numerous typos and issues in the appendix references, which require a careful reread and thorough revision of the paper. For instance:

(Line 161):  *Remark. Detailed mathematical formulation of Wasserstein distance, Sinkhorn regularization, and computational complexity analysis are provided in the Appendix for completeness*: it  lacks an appendix reference.

(Line 269)*The ablation of contribution of head fine-tuning( 5 and OT mask are shown at the appendix.)* :  contains grammatical and formatting errors.

(Line 302)“concrete subspace learning”: This  term is used without any supporting reference or explanation. What does it represent?

(Line 375) *We conduct similar analyses on language tasks using the Flan-T5-base model. (Appendix)*: Table 4 should refer explicitly to Table 4 in Appendix X, where X is the appendix section.

(Line 430) *while additional t-SNE plots 6, 7, 8, 9, 10, for other methods are provided in the appendix*: It does not reference the appendix.

- There are duplicate entries in the bibliography (e.g., Task Arithmetic appears twice)

- The ablation studies (Figure 5(a) and (b) in Appendix A.2) must  be included in the main paper, as they are crucial for understanding the impact of some of the components in the proposed methodology.

Moreover, I have several concerns regarding the methodology, computational cost required by the proposed method, the ablations on the proposed approach and the unclear experimental setting used by the authors.  In the following more details:

2) The method requires a significantly large supervised labeled set for training the classification head, which is not needed by the other competing methods. (see the question section for details)


3) By examining Algorithm 1, it appears that the proposed approach introduces several computational complexities compared to the baselines, while the Complexity Analysis section fails to accurately describe the true computational cost of the method. I believe the approach incurs a significant computational burden, which makes it difficult to assess its applicability in realistic scenarios. The authors’ claim of a “small overhead” compared to training-free approaches is not supported by either the complexity analysis or any experimental results, which are absent from the paper  (see the question section for more details).

4) The criterion for choosing the merging coefficient $\alpha$ is purely empirical, with its value fixed to 0.8 (as shown in the appendix) since it provides the best performance (see the question section for more details).

5) The authors propose the Sinkhorn loss, but its importance within the overall method is not clearly demonstrated. In the paper, all comparisons are made using the full combination of Sinkhorn loss + Mask training + Classifier Tuning (both proposed by the authors) against Adamerging, which learns a per-task $\lambda$ parameter using an unsupervised loss. The ablation in the appendix solely shows the importance of Mask Training and Head Tuning.  Other continual regularization techniques for mitigating distribution shifts are not explored to understand the importance of the Sinkhorn loss (see the question section for more details)

6) The Mask training importance is not sufficiently ablated in the appendix and it is only compared with the experiment when it is deactivated (see questions section for details)

7) In the paper, it is unclear where the fine-tuned models are taken from, and no references are provided. The authors also do not specify how the data is split between the unlabeled set used for the Sinkhorn loss and the labeled set used for supervised learning (see question section for details)

**Questions:**

Q.2) Can the supervised loss be replaced with the Adamerging unsupervised entropy loss on the same unlabeled set used for the Synkhorn loss?

Q.3) In the complexity analysis, the authors claim that there is only a “small overhead” compared to training-free approaches, but this statement appears to be inaccurate and is not supported by the provided analysis or experiments.  The proposed method requires three forward passes per image—one through the current model, one through the previous model, and one through the previously merged model—as well as multiple iterations of Sinkhorn optimization for each epoch. In contrast, training-free approaches only need to validate the merging parameter on a validation set using a single inference step and perform simple operations on the weights. What are the empirical computational times in comparison? Moreover, how computationally expensive is the proposed method relative to Adamerging and OPCM, the closest competitors?

Q.4) Can $\alpha$ be determined automatically based on the characteristics of the masks? In OPCM, the authors show that an adaptive scaling factor can be designed for the task vectors. Could a similar mechanism be applied here?

Q.5) There is extensive literature in continual learning on mitigating distribution shift, including approaches that leverage unlabeled datasets. Simple baseline comparisons such as Learning Without Forgetting (LwF) [1], which applies knowledge distillation only on the current task data (as implemented in FACIL [2]), or PodNet Distillation [3], which closely resembles the regularization loss in Eq. 1, are not evaluated in the paper. Although these methods are relatively old, they remain strong baselines for assessing the effectiveness of the proposed loss and are significantly more efficient than the Sinkhorn loss. Moreover, several recent regularization-based continual learning methods that regularize distribution shifts could further help in understanding the performance of the proposed regularizer (see the survey  PyCIL [4] and the survey 5]).

Q.6) What are the performance when a layerwise scaling factor or taskwise learning factor are learned (as in Adamerging) when the Sinkhorn loss is employed?

Q.7) What dataset splits are used for performing the continual model merging training?  The unlabeled set is a portion of the evaluation set as in Adamerging? Is the labeled set a part of the training set not used for evaluation ?

I believe that the current work represents a preliminary version that could potentially be suitable for future conferences, but it still requires substantial improvements. I encourage the authors to enhance the overall quality of the paper by clarifying the methodology, better motivating the proposed improvements, and strengthening the experimental analysis. In particular, the method should be refined with a stronger focus on efficiency and more comprehensive ablations. For the current submission, my recommendation is rejection.

[1] Z. Li and D. Hoiem, Learning Without Forgetting, IEEE Transactions on Pattern Analysis and Machine Intelligence (TPAMI), 2018.

[2] Masana et al., Class-Incremental Learning: Survey and Performance Evaluation on Image Classification, IEEE Transactions on Pattern Analysis and Machine Intelligence (TPAMI), 2023.

[3] A. Douillard et al., PodNet: Pooled Outputs Distillation for Small-Tasks Incremental Learning, European Conference on Computer Vision (ECCV), 2020.

[4] Zhou et al., Class-Incremental Learning: A Survey, IEEE Transactions on Pattern Analysis and Machine Intelligence (TPAMI), 2024.

[5] Yang et al., Continual Learning: A Systematic Literature Review, Neural Networks, 2025.

---

### Official Review · Reviewer_Piro · 2025-10-31

**Soundness:** 2
**Presentation:** 2
**Contribution:** 2
**Rating:** 2
**Confidence:** 4

**Summary:**

Paper proposes a method for continual fusion of models. The method is based on optimal transport. OTMF (Optimal Transport-based Masked Fusion) is a model-merging framework that aligns task-specific models in feature space using optimal transport. The optimal transport is applied to the output features of the models. They optimize two masks that optimize the alignment with the pre and post training features. Important is that this paper considers continual merging, where two models are merged at each step. Results show that the method outperforms several baselines.

**Strengths:**

- the usage of distribution alignment with optimal transport is new in the context of model merging.
- resulsts show improvements over several baselines on vision (Results on language (table 3) are significantly weaker)

**Weaknesses:**

- The presentation of the paper can be much improved. The authors have included all the components of their method in Sections 3 and 4, but much of the work is left to the reader to connect them (using for example the algorithm). The actual use of the loss described in Section 3.2 is not discussed in the context of optimizing the masks introduced later in Section 4.2. The alternating nature of the mask optimization is mentioned only in Algorithm 1, but it should also be discussed explicitly in the main text. Furthermore, the authors use unclear language in the introduction: the meaning of "feature distribution" is not clear at that point (it becomes clearer later), and the term "semantic geometry" remains undefined.

-The motivation for adopting the continual setting is weak, as it focuses primarily on memory considerations. The authors could strengthen this section by drawing on additional motivations from the continual learning literature, such as privacy concerns. Furthermore, it would be good to compare with other continual merging methods, for instance “Weighted Ensemble Models Are Strong Continual Learners” (ECCV 2024).

- Results of the language models is weaker, do the authors have any explanations for that ? These should be moved to the main paper.

- Given that the authors perform continual merging, the order of fussion becomes important. It would be good to see results for some other order of datasets. Also where did the authors get the applied order from ? It seems different from the standard order.

- given that the authors perform continual merging, one would expect some analysis in terms of stability versus plasticity.

- The method TSVM is not reported in the joint merging setting (Gargiulo, CVPR, 2025). Why is that ? The paper reports much higher results (86% with ViT B-32 for Table 2)?

minor
- Sinkhorn needs citation and/or exlpanation when first mentioned.

**Questions:**

Please address the questions in the weaknesses.

---

### Meta-Review · Area_Chair_earr · 2026-01-06

**Summary:**

1. All reviewers point out written and presentatioin problems, including unclear methodology, fragmented structure, and numerous grammatical errors. The article may need a thorough revision.
2. The motivation needs further justification. The motivation for the continual setting lacks depth, and key components like the Sinkhorn loss are inadequately explained in the main text.
3. There are also many concerns about experiments. Including reliance on labeled data (uncommon in model merging), insufficient ablations, missing per-task results, and unverified computational efficiency claims.

**Reviewer Concerns:**

Since there is no rebuttal, no concerns are addressed.
Concerns about written and presentation problems, motivation expression, and experiments are still outstanding.

**Reviewer Scores:**

Since there is no rebuttal, no scores may be changed.
So the scores may be 2,2,4. I think the scores are below the acceptance threshold.

---

### Decision · Program_Chairs · 2026-01-26

Reject